# Chest Compression-Related Flail Chest Is Associated with Prolonged Ventilator Weaning in Cardiac Arrest Survivors

**DOI:** 10.3390/jcm11082071

**Published:** 2022-04-07

**Authors:** Kevin Kunz, Sirak Petros, Sebastian Ewens, Maryam Yahiaoui-Doktor, Timm Denecke, Manuel Florian Struck, Sebastian Krämer

**Affiliations:** 1Medical Intensive Care Unit, University Hospital Leipzig, 04103 Leipzig, Germany; sirak.petros@medizin.uni-leipzig.de; 2Department of Diagnostic and Interventional Radiology, University Hospital Leipzig, 04103 Leipzig, Germany; sebastian.ewens@gmail.com (S.E.); timm.denecke@medizin.uni-leipzig.de (T.D.); 3Medical Faculty, Institute for Medical Informatics, Statistics and Epidemiology, University of Leipzig, 04107 Leipzig, Germany; maryam.yahiaoui@imise.uni-leipzig.de; 4Department of Anesthesiology and Intensive Care Medicine, University Hospital Leipzig, 04103 Leipzig, Germany; manuelflorian.struck@medizin.uni-leipzig.de; 5Department of Visceral, Transplant, Thoracic and Vascular Surgery, Division of Thoracic Surgery, University Hospital Leipzig, 04103 Leipzig, Germany; sebastian.kraemer@medizin.uni-leipzig.de

**Keywords:** cardiopulmonary resuscitation, chest wall injury, flail chest, ventilator weaning, surgical rib stabilization

## Abstract

Chest compressions during cardiopulmonary resuscitation (CPR) may be associated with iatrogenic chest wall injuries. The extent to which these CPR-associated chest wall injuries contribute to a delay in the respiratory recovery of cardiac arrest survivors has not been sufficiently explored. In a single-center retrospective cohort study, surviving intensive care unit (ICU) patients, who had undergone CPR due to medical reasons between 1 January 2018 and 30 June 2019, were analyzed regarding CPR-associated chest wall injuries, detected by chest radiography and computed tomography. Among 109 included patients, 38 (34.8%) presented with chest wall injuries, including 10 (9.2%) with flail chest. The multivariable logistic regression analysis identified flail chest to be independently associated with the need for tracheostomy (OR 15.5; 95% CI 2.77–86.27; *p* = 0.002). The linear regression analysis identified pneumonia (β 11.34; 95% CI 6.70–15.99; *p* < 0.001) and the presence of rib fractures (β 5.97; 95% CI 1.01–10.93; *p* = 0.019) to be associated with an increase in the length of ICU stay, whereas flail chest (β 10.45; 95% CI 3.57–17.33; *p* = 0.003) and pneumonia (β 6.12; 95% CI 0.94–11.31; *p* = 0.021) were associated with a prolonged duration of mechanical ventilation. Four patients with flail chest underwent surgical rib stabilization and were successfully weaned from the ventilator. The results of this study suggest that CPR-associated chest wall injuries, flail chest in particular, may impair the respiratory recovery of cardiac arrest survivors in the ICU. A multidisciplinary assessment may help to identify patients who could benefit from a surgical treatment approach.

## 1. Introduction

According to the current Advanced Life Support guidelines, cardiopulmonary resuscitation (CPR) requires a chest compression depth of 5–6 cm, at a rate of 100–120 compressions per minute [1,2]. This may result in iatrogenic thoracic injuries, which have been commonly identified in studies of postmortem findings, whereas fatal injuries have not been found [3,4,5,6,7,8]. 

The European Resuscitation Council guidelines for post-resuscitation care and the European Society of Cardiology position paper on cardiac arrest centers do not give recommendations regarding the management of CPR-associated thoracic injuries, other than to perform chest radiography to identify possible pneumothorax [2,9]. There are neither studies to reveal whether thoracic injuries in CPR survivors influence the clinical course, nor if the management of these injuries may lead to an improvement in weaning from the ventilator.

Recently, several promising case series on the use of surgical stabilization of CPR-associated rib fractures have been published [10,11,12]. To summarize our own experience, and as a result of the paucity of reported data, this study aimed to analyze the consequences of CPR-related thoracic injuries on the course of respiratory recovery, during intensive care unit (ICU) treatment in CPR survivors.

## 2. Materials and Methods

In this study, ICU survivors of non-traumatic cardiac arrests were analyzed retrospectively, based on the medical records of the University Hospital of Leipzig, Germany. The analysis included patients who were treated in the medical ICU after CPR between January 2018 and June 2019. The patients who died during their hospital stay were excluded. 

Demographic and clinical data were obtained from the patients’ charts and digital health records. The chest radiography and computed tomography (CT) scans, performed after CPR, were reassessed by a board-certified radiologist, with a focus on thoracic injuries (MEDOS RIS version 9.3.3008, Nexus MagicWeb version VA60C_0115, Visage Imaging, PACS: syngo.plaza, Siemens Healthcare, Erlangen, Germany). The taxonomy of chest injuries was carried out according to the Chest Wall Injury Society collaborators [13]. “Flail segment” described three or more ribs fractured in two or more places. The term “anterior flail segment” was used if a minimum of three rib or costal cartilage fractures were detected on both sides, while “flail chest” described the paradoxical motion observed during clinical examination.

The statistical analysis included tests for the normal distribution of metric variables, such as the Shapiro–Wilk test. The metric data are presented either with mean and standard deviation or with median and 25% and 75% quartiles, based on their distributions. The metric data of the groups were compared with the Mann–Whitney U test, while the categorical data were compared by means of the chi-square test and Fisher’s exact test. A logistic regression analysis was performed to identify the independent predictors of the need for tracheostomy, and a linear regression analysis with analysis of variance (ANOVA) was conducted to identify the predictors of the length of ICU stay and the duration of mechanical ventilation. Either odds ratios (ORs) or beta weights with 95% confidence intervals (CIs) were provided, and a *p*-value < 0.05 was considered to be statistically significant. The statistical analysis was performed using SPSS for Windows version 25 (IBM, Armonk, NY, USA).

## 3. Results

From a total of 316 patients who were treated in the medical ICU after CPR, 111 patients met the inclusion criteria, of which 109 had complete data and were the subjects of the study (Figure 1).

The demographic characteristics of the patients and the radiologically detected injuries are presented in Table 1 and Table 2, respectively. Chest radiography was performed on 76 patients (69.7%), while a thoracic CT scan was performed on 44 patients (40.4%). 

The median CPR duration was significantly longer among patients with out-of-hospital cardiac arrest (OHCA) than among those with in-hospital cardiac arrest (IHCA) (14 vs. 2.5 min; *p* < 0.001). The mean number of rib fractures in the patients with at least one detected fracture was 6.7 ± 3.7. Flail chest was detected in 10 patients (9.2%), while an anterior flail segment was observed in 15 patients (16.5%). Pneumothorax was observed in three patients, and there were no cases of hemothorax. Sternal fractures or an anterior flail segment were more frequent among patients with OHCA, compared to those with IHCA (*p* = 0.048).

Seventy-six (69.7%) of the included patients were intubated (78% of the OHCA group) and mechanically ventilated during CPR. Seventy-one patients (93.4%) were successfully weaned from the ventilator during their ICU stay, while four patients remained mechanically ventilated until they were transferred to a rehabilitation unit. Due to extubation failure, 21 (19.3%) patients underwent percutaneous dilatational tracheostomy to facilitate weaning from mechanical ventilation. 

The patients with flail chest had a significantly longer CPR duration, longer length of ICU stay, more days on mechanical ventilation, and higher rates of tracheostomy and pneumonia (Table 3). 

The multivariable logistic regression analysis (Nagelkerke R-squared 0.321; Hosmer–Lemeshow *p* = 0.545) showed that flail chest was an independent predictor of the need for tracheostomy (OR 15.5; 95% CI 2.77–86.72; *p* = 0.002) (Table 4).

Furthermore, the multivariable linear regression analysis (ANOVA *p* < 0.001; adjusted R-Square 0.363) showed that pneumonia (β 11.34; 95% CI 6.70–15.99; *p* < 0.001) or the detection of multiple rib fractures (β 5.97; 95% CI 1.01–10.93; *p* = 0.019) independently influenced the length of ICU stay (Table 5).

Another multivariable linear regression analysis (ANOVA *p* < 0.001; adjusted R-Square 0.290) showed that flail chest (β 10.45; 95% CI 3.57–17.33; *p* = 0.003) and pneumonia (β 6.12; 95% CI 0.94–11.31; *p* = 0.021) significantly prolonged the duration of mechanical ventilation (Table 6).

Four survivors failed to show adequate progression in weaning from the ventilator after receiving a tracheostomy, for a number of reasons, including flail chest. In a case-by-case decision, the treating physician, together with a thoracic surgeon, identified patients who would undergo surgical stabilization of their rib fractures, who were eventually weaned from mechanical ventilation (Table 7; Figure 2). 

## 4. Discussion

The results of this study suggest that the CPR-associated thoracic injury pattern among survivors of non-traumatic cardiac arrest may prolong the weaning process from the ventilator and ICU stay.

Rib fractures were the most frequently detected thoracic bone damage in our study. Most previous studies on injuries after CPR utilized necropsy, postmortem CT scans, or both, with the primary aim of demonstrating the differences between injuries caused by manual chest compressions and automated chest compression devices [3,4,6,8]. A recently published study, regarding CPR-associated thoracic injuries, included patients after OHCA, who received chest CT imaging after CPR. The median number of rib fractures was comparable with our results. An association between the detection of rib fractures and a prolonged length of ICU stay was found in the patients, which supports our results [12].

The frequency of clinically detected flail chest in our cohort was almost twice as high as that reported in a previous publication that was conducted almost half a century ago, which could be the result of higher survival rates [14]. The patients with flail chest had a higher rate of pneumonia, longer ICU stay and mechanical ventilation, and higher tracheostomy rates, compared with patients without flail chest. On the other hand, pneumonia and the detection of a rib fracture prolonged the overall length of ICU stay, while flail chest did not. Taken together, rib fractures, and especially multiple rib fractures that lead to flail chest, seem to influence post-resuscitation care in the ICU.

Regarding the treatment options after CPR-associated chest injuries, the present study was not designed to confirm whether early surgical rib fixation could influence the outcome. In this study, we present a series of four cases, in which surgical rib fractures might have contributed to improved respiratory function.

Recent data have shown improvements in the numeric pain score of patients with multiple rib fractures after surgical stabilization [15]. The Eastern Association for the Surgery of Trauma recommends rib fixation in patients with flail chest, to reduce the need for tracheostomy, increase the number of ventilator-free days, and reduce the length of ICU and hospital stays [16]. Whether patients with traumatic flail chest clearly benefit from surgical therapy remains controversial. Overall, surgical interventions have shown promising long-term outcomes and low complication rates [17,18,19]. However, these data refer to traumatic chest injuries and reflect a different patient cohort, in comparison to survivors of cardiac arrest with specific patterns of injuries. In a study of 61 ventilator-dependent patients with flail chest (including 7 CPR survivors), surgical rib fixation was associated with a shortened ICU stay and improved cost effectiveness [20].

Due to a considerable lack of data regarding the effect of surgical rib fixation on the course of the weaning process, more research is urgently needed. Future research should provide evidence of whether this procedure will remain a rare treatment option, or if it has the potential to change clinical practice patterns. This could particularly affect hospitals without cardiothoracic surgery departments, and the referral and capacity management of certified cardiac arrest centers.

As a result of this study, we have introduced a multidisciplinary assessment of CPR survivors with promising rehabilitation potential and a relevant probability of flail chest-related weaning failure, and whether they might benefit from surgical rib fixation. This assessment includes intensivists, respiratory therapists and thoracic surgeons.

### Limitations

Retrospective single-center studies, with a limited number of patients, may result in selection bias. Less than half of the included patients underwent chest CT after CPR, which may have led to diagnostic bias, because the diagnostic accuracy of plane chest radiography for bone lesions is lower than CT [21]. Furthermore, deceased patients were not included in this study, who may have presented with other injury characteristics and different results.

## 5. Conclusions

Our results suggest that CPR-related injuries may impair the respiratory recovery of cardiac arrest survivors. The presence of flail chest may be associated with the need for tracheostomy, prolonged mechanical ventilation and pneumonia, whereas multiple rib fractures and pneumonia may contribute to longer ICU stays. A multidisciplinary assessment, including intensivists, respiratory therapists and thoracic surgeons, may help to identify the patients who may benefit from surgical rib fixation and chest stabilization, which has to be confirmed in future studies.

## Figures and Tables

**Figure 1 jcm-11-02071-f001:**
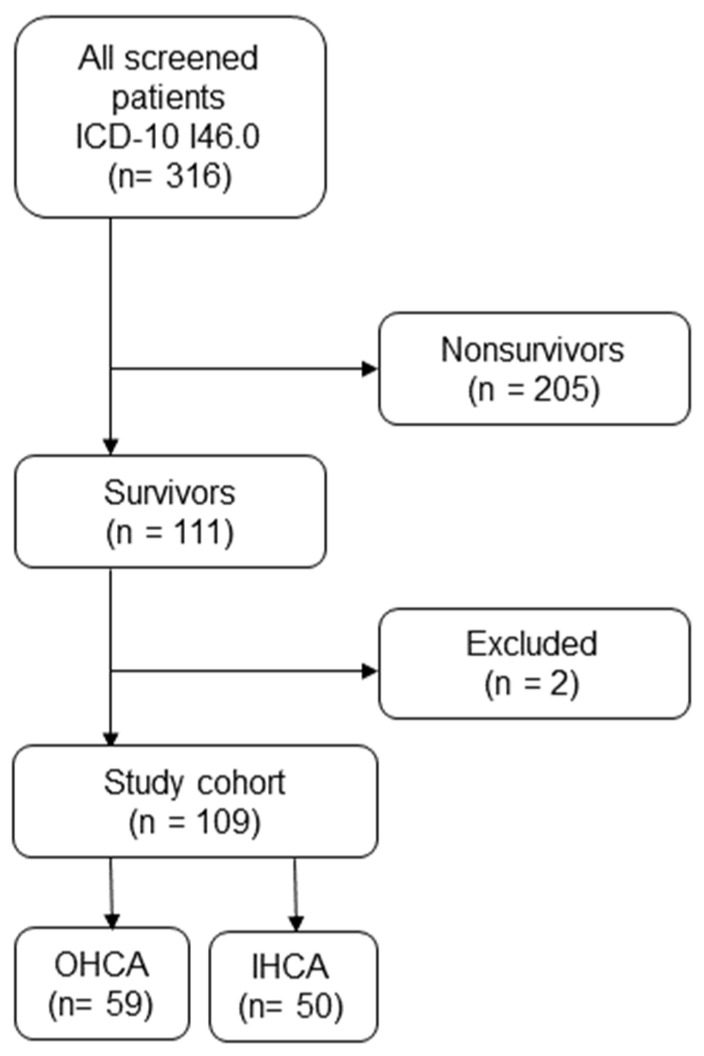
Study flowchart.

**Figure 2 jcm-11-02071-f002:**
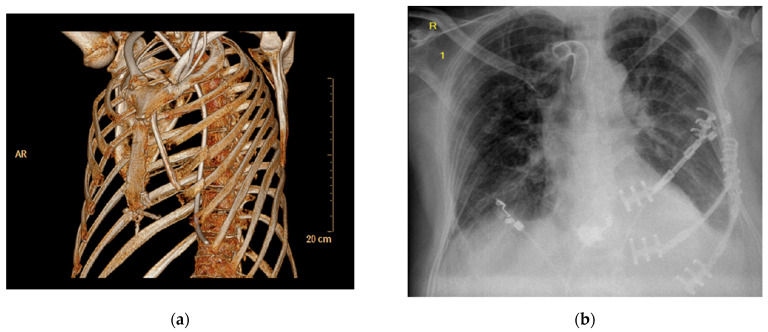
Chest CT reconstruction of a patient with multiple rib fractures and flail chest after 65 min of CPR, including the use of extracorporeal CPR, due to massive pulmonary embolism. (**a**) The patient underwent surgical rib fixation for stabilization (**b**) and was finally weaned from the ventilator after 37 days of mechanical ventilation.

**Table 1 jcm-11-02071-t001:** Demographic characteristics of 109 patients with CPR-related chest injuries.

Variable	Value
OHCA, *n* (%)	59 (54.1%)
Male, *n* (%)	67 (61.5%)
Age, years	69 (56–77)
Height, cm	173 (165–180)
Weight, kg	80 (71–85)
BMI, kg/m^2^	26.24 (23.60–29.31)
Heart failure, *n* (%)	44 (40.4%)
COPD, *n* (%)	15 (13.8%)
Pneumonia, *n* (%)	55 (50.5%)
eCPR, *n* (%)	2 (1.8%)
ACCD, *n* (%)	7 (6.4%)
CPR duration, min	5 (2.25–15)
ICU LOS, days	7 (3–18)
Mechanical ventilation, days	5 (2–14)
Tracheostomy, *n* (%)	21 (19.3%)

Data are presented as numbers (percentages) or medians (IQR). CPR, cardiopulmonary resuscitation; BMI, body mass index; COPD, chronic obstructive pulmonary disease; eCPR, extracorporeal CPR; ACCD, automated chest compression device; OHCA, out-of-hospital cardiac arrest; ICU LOS, intensive care unit length of stay.

**Table 2 jcm-11-02071-t002:** CPR-related thoracic injuries, detected by thoracic CT imaging or chest radiography.

	Number (%)
Rib fractures	34 (31.2%)
Unilateral	9 (8.3%)
Bilateral	25 (22.9%)
Anterior flail segment	18 (16.5%)
Sternal fracture	15 (13.8%)
Hemothorax	0
Pneumothorax	3 (2.8%)
No relevant chest injuries	71 (65.1%)

CPR, cardiopulmonary resuscitation; CT, computed tomography.

**Table 3 jcm-11-02071-t003:** Characteristics of patients with and without CPR-related flail chest.

Variable	Flail Chest (*n* = 10)	No Flail Chest (*n* = 99)	*p*-Value
Male	9 (90%)	58 (58.6%)	0.048
Age, years	64.5 (61–68.25)	69 (55–77)	0.862
Height, cm	180 (170–185)	172 (165–180)	0.580
Weight, kg	80 (75–94.75)	79 (70–85)	0.236
BMI, kg/m^2^	26.86 (24.64–29.24)	26.24 (23.38–29.39)	0.702
Heart failure, *n* (%)	4 (40%)	40 (40.4%)	0.628
COPD, *n* (%)	1 (10%)	14 (14.1%)	0.586
Pneumonia, *n* (%)	10 (100%)	45 (45.5%)	0.001
eCPR, *n* (%)	1 (10%)	1 (1%)	0.176
ACCD, *n* (%)	1 (10%)	6 (6.1%)	0.500
CPR duration, min	19 [10,11,12,13,14,15,16,17,18,19,20,21]	5 (2–15)	0.009
ICU LOS, days	25 (19.5–33.75)	6 (3–15)	<0.001
Mechanical ventilation, days	18.5 (14–28)	4 (2–10)	0.001
Tracheostomy, *n* (%)	8 (80%)	13 (13.1%)	<0.001

Date are presented as numbers (percentages) or medians (IQR). BMI, body mass index; COPD, chronic obstructive pulmonary disease; ACCD, automated chest compression device; CPR, cardiopulmonary resuscitation; eCPR, extracorporeal CPR; ICU LOS, intensive care unit length of stay.

**Table 4 jcm-11-02071-t004:** Associations with the need for tracheostomy in logistic regression analysis.

Variable	Univariable OR (95% CI)	*p*-Value	Multivariable OR (95% CI)	*p*-Value
Age	1.01 (0.97–1.04)	0.783		
BMI	1.00 (0.93–1.07)	0.950		
Heart failure	1.16 (0.42–2.99)	0.831		
COPD	2.44 (0.73–8.01)	0.146		
Pneumonia	-	0.997		
Flail chest	26.46 (5.05–138.56)	<0.001	15.50 (2.77–86.72)	0.002
Duration of CPR	1.03 (1.00–1.06)	0.04	1.02 (0.98–1.05)	0.339
Rib fractures	5.19 (0.97–1.04)	0.001	0.36 (0.11–1.13)	0.080
Anterior flail segment	0.40 (0.13–1.22)	0.106		

OR, odds ratio; CI, confidence interval; BMI, body mass index, COPD, chronic obstructive pulmonary disease; CPR, cardiopulmonary resuscitation.

**Table 5 jcm-11-02071-t005:** Associations with the length of ICU stay in linear regression analysis.

Variable	Univariable β (95% CI)	*p*-Value	Multivariable β (95% CI)	*p*-Value
Age	−0.69 (3.57–17.33)	0.442		
BMI	0.33 (0.94–11.31)	0.110		
Heart failure	1.40 (−0.34–9.44)	0.604		
COPD	7.46 (3.57–17.33)	0.050		
Pneumonia	14.99 (0.94–11.31)	<0.001	11.34 (6.70–15.99)	<0.001
Flail chest	17.81 (−0.34–9.44)	<0.001	7.85 (−0.59–15.76)	0.052
Duration of CPR	0.27 (3.57–17.33)	0.040	0.06 (−0.10–0.22)	0.489
Rib fractures	11.84 (0.94–11.31)	<0.001	5.97 (1.01–10.93)	0.019
Anterior flail segment	6.09 (−0.34–9.44)	0.086		

β, beta weight; CI, confidence interval; BMI, body mass index; COPD, chronic obstructive pulmonary disease; CPR, cardiopulmonary resuscitation.

**Table 6 jcm-11-02071-t006:** Associations with mechanical ventilation in linear regression analysis.

Variable	Univariable β (95% CI)	*p*-Value	Multivariable β (95% CI)	*p*-Value
Age	0.16 (−0.02–0.33)	0.750		
BMI	−0.09 (−0.53–0.35)	0.690		
Heart failure	−0.71 (−6.03–4.62)	0.791		
COPD	3.30 (−3.68–10.27)	0.349		
Pneumonia	9.17 (3.67–14.71)	0.01	6.12 (0.94–11.31)	0.021
Flail chest	14.58 (7.87–21.8)	<0.001	10.45 (3.57–17.33)	0.003
Duration of CPR	0.10 (−0.07–0.27)	0.236		
Rib fractures	8.35 (3.32–13.39)	0.001	4.55 (−0.34–9.44)	0.068
Anterior flail segment	2.44 (−3.70–8.58)	0.431		

β, beta weight; CI, confidence interval; BMI, body mass index; COPD, chronic obstructive pulmonary disease; CPR, cardiopulmonary resuscitation.

**Table 7 jcm-11-02071-t007:** Case characteristics of patients who underwent surgical rib fixation.

Variable	Patient 1	Patient 2	Patient 3	Patient 4
Age	61 years	53 years	61 years	66 years
Sex	male	male	male	male
BMI [kg/m^2^]	24.7	24.7	29.2	22.5
Reason for CPR	Pulmonary embolism	Myocardial infarction	Myocardial infarction	Hypoxia
Relevant comorbidities	Aspiration pneumonia	Aspiration pneumonia	Influenza pneumonia Acinetobacter pneumonia	Aspiration pneumonia Heart failure COPD
Duration of CPR	65 min	20 min	15 min	21 min
Length of ICU stay	45 days	27 days	69 days	18 days
Mechanical ventilation	37 days	18 days	59 days	16 days
Number of fractured ribs	9 ribs	8 ribs	7 ribs	14 ribs
Anterior flail segment	Yes	Yes	Yes	Yes

BMI, body mass index; CPR, cardiopulmonary resuscitation; ICU, intensive care unit; COPD, chronic obstructive pulmonary disease.

## Data Availability

The data supporting the findings of this study are available from the corresponding author, upon reasonable request.

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
