# Peer review of "Chest Compression-Related Flail Chest Is Associated with Prolonged Ventilator Weaning in Cardiac Arrest Survivors"

_jcm, 2022, doi:10.3390/jcm11082071_

Round 1

Reviewer 1 Report

I commend the authors for undertaking the present study providing information about chest compression-related injuries and their associated outcomes in cardiac arrest survivors.

It is a well-written study. The methods are clearly described and the results are presented in a clear and comprehensible way.

However, I feel that the findings described (flail chest is associated with tracheostomy/prolonged mechanical ventilation and pneumonia in cardiac arrest survivors; multiple rip fractures and pneumonia contribute to longer ICU stays) are quite obvious.

The clinically interesting question would be who in this cohort of patients (multiple rib fractures after CPR in non traumatic cardiac arrest) would benefit from surgical rib fixation.

Author Response

RESPONSE: We would like to thank the reviewer for his comments. We agree with the reviewer that it is important to identify patients who might benefit from surgical rib fixation. We have added to the discussion section:

“Due to considerable lack of data regarding the effect of surgical rib fixation on the course of the weaning process, more research is urgently needed. Future research should provide evidence whether this procedure will remain a rare treatment option or if it has the potential to change clinical practice patterns. This could particularly affect hospitals without cardiothoracic surgery departments and the referral and capacity management of certified cardiac arrest centers.

As the result of this study, we have introduced an multidisciplinary assessment of CPR-survivors with promising rehabilitation potential and a relevant probability of flail chest-related weaning failure whether they might benefit from surgical rib fixation. This assessment includes intensivists, respiratory therapists and thoracic surgeons.”

Reviewer 2 Report

I think this is a good paper, which has obviously been years in the making. I think the introduction, methods, and results section are well written, and do not hide behind large amount of stats. I think the issue is prescient as there are some trials going on at the moment looking at rib fractures and whether fixation might help. The main issue is how to infer conclusions- always more research is needed and how to incorporate this outside of large trauma units with no access to cardiothoracics. Also how have the authors changed their practice after this? Do they now offer surgery to all comers with flail chests? 

Author Response

RESPONSE: We would like to thank the reviewer for his comments. We agree with the raised points and included them in the discussion section:

“Due to considerable lack of data regarding the effect of surgical rib fixation on the course of the weaning process, more research is urgently needed. Future research should provide evidence whether this procedure will remain a rare treatment option or if it has the potential to change clinical practice patterns. This could particularly affect hospitals without cardiothoracic surgery departments and the referral and capacity management of certified cardiac arrest centers.

As the result of this study, we have introduced an multidisciplinary assessment of CPR-survivors with promising rehabilitation potential and a relevant probability of flail chest-related weaning failure whether they might benefit from surgical rib fixation. This assessment includes intensivists, respiratory therapists and thoracic surgeons.”

Round 2

Reviewer 1 Report

Unfortunately, the additional sentences added to the discussion cannot reduce my concerns about the value and the originality of the present study.

As mentioned, I believe that the findings described are somehow obvious and do not warrant publication in this high impact journal. The clinically important question of who would benefit from rib fixation after CPR related rib fractures cannot be answered with the present study design.